# Pseudocapacitance-Enhanced Storage Kinetics of 3D Anhydrous Iron (III) Fluoride as a Cathode for Li/Na-Ion Batteries

**DOI:** 10.3390/nano12224041

**Published:** 2022-11-17

**Authors:** Tao Zhang, Yan Liu, Guihuan Chen, Hengjun Liu, Yuanyuan Han, Shuhao Zhai, Leqing Zhang, Yuanyuan Pan, Qinghao Li, Qiang Li

**Affiliations:** College of Physics, Weihai Innovation Research Institute, College of Materials, Qingdao University, Qingdao 266071, China

**Keywords:** full-cell, kinetic analysis, metal fluorides, storage mechanism

## Abstract

Transition metal fluoride (TMF) conversion cathodes, with high energy density, are recognized as promising candidates for next-generation high-energy Li/Na-ion batteries (LIBs/SIBs). Unfortunately, the poor electronic conductivity and detrimental active material dissolution of TMFs seriously limit the performance of TMF-LIBs/SIBs. A variety of FeF_3_-based composites are designed to improve their electrochemical characteristics. However, the storage mechanism of the conversion-type cathode for Li^+^ and Na^+^ co-storage is still unclear. Here, the storage mechanism of honeycomb iron (III) fluoride and carbon (FeF_3_@C) as a general cathode for LIBs/SIBs is analyzed by kinetics. In addition, the FeF_3_@C cathode shows high electrochemical performance in a full-cell system. The results show that the honeycomb FeF_3_@C shows excellent long-term cycle stability in LIBs (208.3 mA h g^−1^ at 1.0 C after 100 cycles with a capacity retention of 98.1%). As a cathode of SIBs, the rate performance is unexpectedly stable. The kinetic analysis reveals that the FeF_3_@C cathode exhibit distinct ion-dependent charge storage mechanisms and exceptional long-durability cyclic performance in the storage of Li^+^/Na^+^, benefiting from the synergistic contribution of pseudocapacitive and reversible redox behavior. The work deepens the understanding of the conversion-type cathode in Li^+^/Na^+^ storage.

## 1. Introduction

Energy shortage and environmental pollution make it urgent to store and use green and sustainable energy [1]. Therefore, rechargeable energy storage systems have broad market prospects and economic benefits [2]. Among them, because lithium-ion batteries (LIBs) have the highest output voltage and large energy density, it has been widely used in electric vehicles, wearable devices, and smart grid as mature energy storage and conversion equipment [3,4]. However, the limitation of lithium salt resources that can be directly used on the earth greatly hinders the utilization of lithium salt in LIBs, especially for large-scale energy storage stations [5,6]. Recently, sodium-ion batteries (SIBs) have been considered the most powerful competitor to replace LIBs [7,8]. Because not only does Na^+^ show similar electrochemical behavior to Li^+^ in the batteries, but also sodium is the smallest, lightest alkali element except lithium and is rich in reserves [9]. Therefore, the development of high-performance cathodes with Li^+^ and Na^+^ storage and transmission is of great significance [6,10,11,12,13].

The conventional intercalation-type cathode materials of LIBs, such as lithium cobalt oxide (LCO) and lithium nickel manganese cobalt oxide (NMC), are closely approached their theoretical limits and further promote their energy density and may compromise cell safety [14]. Therefore, the breakthrough in performance needs to develop a new concept in material research. Transition metal fluorides (TMFs) release all possible redox reactions during lithiation and de-lithiation to transfer two or three electrons with each metal atom/ion, which seems to provide an important reference option for obtaining a larger specific capacity [14,15]. Furthermore, compared with the intercalation materials, the mechanism of the conversion reaction between the electrode and alkali metal does not depend on the size of alkali metal ions, so it has better cycle stability in theory [12]. As a representative of TMFs with great application prospects, iron (III) fluoride (FeF_3_) has an average potential of 2.74 V and a volumetric capacity of up to 2196 mAh cm^−3^ [16]. The high theoretical capacity of 237 mAh g^−1^ is maintained in a high operating voltage range (2.0–4.5 V) even under the single electron transfer of FeF_3_ in the LIBs [17,18,19]. Unfortunately, the wide band gap of FeF_3_ exhibits poor electronic conductivity due to the high ionic strength of the metal fluoride bond. Another problem is the electrode pulverization and dissolution caused by volume change during charging and discharging [20,21,22,23,24]. According to previous studies, nanoscale and carbon composite designs are very efficient strategies for mitigating these problems [17,21,25,26,27,28], which can produce point defects and shorten the Li^+^/Na^+^ diffusion distance, thus delivering high conductivity and Li^+^/Na^+^ storage [29,30]. However, few works have focused on the storage mechanisms of the FeF_3_ based on composites.

In this work, 3D anhydrous iron (III) fluoride and carbon (FeF_3_@C) were fabricated to study the storage mechanisms and the energy characteristics of the FeF_3_ cathode in the LIBs and SIBs. Dynamic measurements revealed that the storage of Li^+^ and Na^+^ by FeF_3_@C of honeycomb architecture resulted from the synergistic effect of conversion reaction and capacitive behavior, and the capacitive storage accounted for the majority. In addition, we realized the electrochemical performance measurements of FeF_3_@C in a full cell, deepening its commercial application value.

## 2. Materials and Methods

### 2.1. Synthesis of Fe_3_C@C Composite

In a typical preparation: 3.0 g of ferric nitrate hydrate (Fe(NO_3_)_3_·9H_2_O) (Aladdin, Shanghai, China) and 1.8 g of polyvinylpyrrolidone (PVP, k90) (Aladdin, Shanghai, China) was slowly added to 100 mL of deionized water and stirred until completely dissolved to obtain an orange-yellow mixed solution. Then the mixed solution was placed in a blast dryer, dried at 90 °C for 12 h, and evaporated to dryness.

Adequate gel composite clusters were placed in alumina ceramic boats and carbonized in an Ar atmosphere in a tubular furnace. With the heating rate of 3 °C min^−1^ to 250 °C holding 1 h, then with the rate of 5 °C min^−1^ to 750 °C holding 2 h, and finally naturally fall to room temperature.

### 2.2. Synthesis of FeF_3_@C Composite

A certain amount of iron carbide was heated to 280 °C for 2 h at a heating rate of 10 °C min^−1^ in mixed gas (Ar/NF_3_, 10% NF_3_) and then cooled naturally.

### 2.3. Material Characterization

The crystal structure of the FeF_3_@C was characterized using X-ray diffraction (XRD) (Bruker, Billerica, MA, USA) with high-intensity Cu Kα radiation. The chemical bonding state was analyzed by X-ray photoelectron spectroscopy (XPS) using a Thermo Scientific ESCALAB 250XI photoelectron spectrometer (Thermo Fisher Scientific, Wuhan, China). The surface morphology was investigated by scanning electron microscope (SEM; JSM-6700F) (JEOL, Beijing, China). Transmission electron microscopy (TEM) and high-resolution transmission electron microscopy (HR-TEM) were carried out by a JEOL 100CX instrument (JEOL, Beijing, China). The porosity of the materials was characterized by using ASAP-2010 (Micromeritics, Shanghai, China).

### 2.4. Electrochemical Characterization

The FeF_3_@C cathode was made from the compositions of active materials (70 wt%), conductive carbon black (super P, 20 wt%), and binder ((polyvinylidene fluoride) (PVDF) (DoDoChem, Suzhou, China) in N-methyl-2-pyrrolidone (NMP) (DoDoChem, Suzhou, China), 10 wt%), forming a slurry. The slurry was evenly coated on the aluminum foil collector, and then the electrode was obtained overnight at 120 °C in a vacuum dryer. The average active mass loading is about 1.2–2.0 mg cm^−2^ on current collectors. The CR2032 coin batteries were assembled in an argon-filled glove box (<0.1 ppm of H_2_O, <0.1 ppm of O_2_). Lithium foils were used as an anode in LIBs (the prelithiation of graphite was used as an anode in the full cells), and polypropylene/polyethylene microporous film (Celgard, Charlotte, NC, USA) as the separator. The 1M LiPF6 in fluoroethylene carbonate/ethyl methyl carbonate (FEC/EMC, 3:7 *v/v*) for the electrolyte. In SIBs, sodium foil (Aladdin, Shanghai, China) and glass microfiber filters (Whatman, Beijing, China) were used as the counter electrode and separators, respectively. A 1.0 M solution of NaClO_4_ in ethylene carbonate/dimethyl carbonate (EC/DEC, 1:1 *v/v*) with 5% fluoroethylene carbonate (FEC) was used as an electrolyte. Charge-discharge tests were conducted by a battery test system (Netware, Shenzhen, China) at various current rates between 2.0 and 4.5 V vs. Li/Li^+^ (Na/Na^+^). Cyclic voltammetry (CV) was carried out on a CHI660E electrochemical workstation (CH Instruments Inc., Shanghai, China). Electrochemical impedance spectroscopy (EIS) was measured at a fully discharged state (~2.0 V) in the frequency range from 100 kHz to 10 mHz with 5 mV amplitude on a CHI660E electrochemical workstation (CH Instruments Inc., Shanghai, China).

## 3. Results and Discussion

Figure 1 is the scheme of the formation process of the FeF_3_@C, and a detailed description of the method is shown in the Experimental Section [26]. The 3D honeycomb Fe_3_C@C is obtained by carbonization of the gel precursor at high-temperature, and the size of the honeycomb channel varies from hundreds of nanometers to several microns, as shown in the field-emission scanning electron microscopy (FESEM) image in Figure 2a. As shown in the FESEM image at high magnification in Appendix A, the Fe_3_C nanoparticles are uniformly embedded on the thin carbon wall. After fluorination with NF_3_ gas, it is worth noting that the composites of FeF_3_@C still maintain the honeycomb morphology with visible honeycomb channels and honeycomb walls (Figure 2b), and the carbon wall surface became rough with the increase of particle (Fe→FeF_3_) size (Appendix A). The nanoparticles embedded in the carbon wall have a strong tolerance to the strain and stress caused by local volume changes, so honeycomb morphology can be maintained during cycling. To get more detailed information on the FeF_3_@C, a high-resolution transmission electron microscopy (HRTEM) was conducted. The lattice spacing of 0.37 nm corresponds to (012) planes of FeF_3_, as shown in Figure 2c. The diffraction rings can be identified as (012), (110), (024) and (122) planes of FeF_3_ from the selected area electron diffraction (SAED) (Figure 2d). In addition, the X-ray diffraction (XRD) results in Figure 2e,f indicate that the PVP/iron nitrate is transformed into Fe_3_C@C composite after high-temperature carbonization and then turned into a single-phase FeF_3_@C composite after low-temperature fluorination [31]. The energy-dispersive X-ray spectroscopy (EDS) elemental mapping results (Figure 2g) further demonstrate that the elements of Fe, F, and C are uniformly distributed in honeycomb FeF_3_@C.

The chemical valence states are noted from the XPS spectrum for Fe, F and C in Figure 3a. In high-resolution XPS (HRXPS) of Fe 2p, two spin-orbit doublets and two shake-up satellites (abbreviated as “Sat.”) peaks are identified in Figure 3b. The signals of Fe 2p_3/2_ (711.4 eV) and Fe 2p_1/2_ (724.9 eV) are attributed to Fe^3+^. The F 1s core-level emission shows a strong peak at 685.3 eV (Figure 3c), which can be assigned to F^−^, while the peaks at 284.6 eV (Figure 3d) are ascribable to the C 1s binding energy, which is useful for energy calibration. All of the results are coincident with the previous XPS data of FeF_3_ [32]. The porous structure is evaluated by nitrogen desorption/adsorption isotherms and further analyzed by Barrett–Joyner–Halenda (BJH) method (Figure 3e). It is revealed that the FeF_3_@C composites have a high surface area of 80.36 m^2^ g^−1^, which is conducive to enhancing the penetration of electrolytes and providing considerable space for volume expansion [33,34,35,36]. The Raman spectrum of FeF_3_@C is demonstrated in Figure 3f. The peaks at 1580 cm^−1^ (G) and 1360 cm^−1^ (D) are related to the graphitic carbon and disordered or defective carbon, respectively. The FeF_3_@C contains both graphitic carbon and amorphous carbon. As shown in the image, the G peak is higher than the D peak, indicating that the material has a high graphitization degree, which greatly improves the ion transport and electronic conductivity of FeF_3_ [37].

To demonstrate the storage performance of the prepared honeycomb FeF_3_@C cathode in Li^+^ cells, half-cells were assembled; their electrochemical performance is characterized in Figure 4. The charge-discharge tests cycled at different current densities (from 0.5 C to 10 C) show considerable rate capabilities. Especially at a high current density of 10 C, the cell still exhibits a capacity of 160 mAh g^−1^, and when returning to 0.5 C, it still maintains a high capacity of 222.7 mAh g^−1^. Similarly, the corresponding galvanostatic charge and discharge curves of FeF_3_@C at different rates show that almost no polarization occurs even at 10 C, indicating that the honeycomb structure endows Li^+^ with a strong diffusion ability, which directly proves the excellent rate capability of LIBs [35,38]. To gain more insights into the electrochemical properties of the FeF_3_@C cathode, the EIS measurements were carried out for LIBs after the initial deep cycle with a fresh CEI film and 100 cycles. The Nyquist plots of LIBs and the equivalent circuit model are shown in Figure 4c. The semicircle in the high-frequency region is related to the charge transfer resistance, and the inclined line in the low-frequency region is tightly correlated with the diffusion of alkali metal ions in the electrode [39,40]. In the equivalent circuit diagram, *R_s_* is considered to be the electrolyte resistance, while *R_ct_* is the charge transfer resistance at the interface. The CPE1 is the constant phase element, which represents the double-layer capacitance and the passivation film capacitance [41]. The slope of the line in the low-frequency region denotes the Warburg impedance (*W_o_*) because of Li^+^ diffusion [42,43]. The impedance curves of the two different test states almost coincide, indicating that the honeycomb structure can always provide efficient charge transfer and Li^+^ transmission, confirming a stable interface between electrode and electrolyte [34]. Besides the studies of rate performance and impedance, the cycle stability of the FeF_3_@C cathode is also investigated. As shown in Figure 4d, the LIBs have excellent capacity stability in 100 cycles at the current density of 1.0 C, showing capacity retention of 98.6% and a stable coulombic efficiency close to 100%.

To further examine the detailed transport kinetic properties of the FeF_3_@C-LIBs, the CV curves (Figure 5a) at different scan rates of 0.4–2.0 mV s^−1^ are applied to evaluate the contributions of diffusion and pseudocapacitive effects. All CV curves exhibit two pairs of redox peaks (P_1O_ and P_2O_) and are nearly entirely reversible, which can be ascribed that the FeF_3_ turns to Li_0.5_FeF_3_ (tri-rutile), and then decomposes to FeF_2_ (rutile) and LiF during lithiation at high voltages between 4.5 and 2.0 V [26]. Besides, with the increase of scanning rate, the peak current increases and the curve shape shows a negligible change. Based on the Randles-Sevcik equation, the peak current corresponding to the redox is linearly fitted with the square root of the scanning rate in Appendix A. Furthermore, the power law formula between the measured current (i) and sweep rates (v) can be expressed as
(1)$$i=av^{b}\;(a\;\mathrm{and}\;b\;\mathrm{are adjustable parameters})$$
which is used to analyze the relationship between peak current and scanning rates [44]. Given Dunn’s empirical formula, the *b*-value of FeF_3_@C can be determined by fitting the straight line of log i versus log v. The fitting lines with slopes of 0.97/0.89 and 0.90/0.90 represent the *b*-value of P_1O_ P_1R_ and P_2O_ P_2R,_ respectively. The *b*-value fitting of all peaks is greater than 0.5 in Figure 5b, indicating that the contribution of pseudocapacitive behavior is prominent in the process of lithiation/de-lithiation [6]. Furthermore, at a stated voltage, the diffusion-controlled part ($k_{1}v^{1/2}$) and a pseudocapacitive fraction ($k_{2}v$) can be quantified abiding the following equation:(2)$$i\left(V\right)=k_{1}v^{1/2}+k_{2}v$$

Equation (2) can be reformulated as:(3)$$i\left(V\right)/v^{1/2}=k_{1}+k_{2}v^{1/2}$$

According to Equation (3), the values of $k_{1}$ and $k_{2}$ can be determined by the linear plot of $i\left(V\right)/v^{1/2}$ as a function of $v^{1/2}$, and then the function relationship between current and voltage in the pseudocapacitive behavior control process can be obtained. The redox pseudocapacitance-like contribution in the FeF_3_@C cathode at 1.6 mV s^−1^ is calculated to be 87.4% (Figure 5c). In Figure 5d, the contribution rate of capacitive increases to 89.8% with the increase of scanning rate at 2.0 mV s^−1^, indicating that the capacitive contribution plays a major role in the entire capacity, especially at high rates.

The galvanostatic intermittent titration technique (GITT) measurement proposed by Weppner et al., as a reliable electrochemical technique to evaluate transport kinetics, has been used to measure the Li^+^ diffusion characteristics of the electrode [45]. The GITT curve of LIBs in the second cycle of lithiation/de-lithiation is shown in Figure 5e. The diffusion coefficient of Li^+^ can be calculated according to Equation (4) [46,47]:(4)$$D_{Li^{+}}=\frac{4}{\pi }{\left(\frac{mV_{M}}{MS}\right)}^{2}{\left(\frac{dE/dx}{dE/d\sqrt{\tau }}\right)}^{2}\left(\tau \ll L^{2}/D_{Li^{+}}\right)$$
where the m (g) is mass, the M (g mol^−1^) and $V_{M}\;$(cm^3^ mol^−1^) are the molecular weight, and molar volume of the active material, respectively, where the $V_{M}$ is about 28.5 cm^3^ mol^−1^ could be calculated according to the content of FeF_3_ in FeF_3_@C [26]. S (cm^2^) means the contact area between the electrode and electrolyte. L refers to the diffusion length of Li^+^. The relationship between the potential (E) and the square root of the relaxation time $\left(\tau^{1/2}\right)$ shows a relatively linear relationship (Appendix A). Therefore, Equation (4) can be further written as Equation (5) [48]:(5)$$D_{Li^{+}}=\frac{4}{\pi }{\left(\frac{mV_{M}}{MS}\right)}^{2}{\left(\frac{∆E_{s}}{∆E_{\tau }}\right)}^{2}$$

The diffusion coefficient of Li^+^ at different potentials was calculated by Equation (5) (Figure 5f). The largest $D_{Li^{+}}$ of the FeF_3_@C electrode is 1.26 × 10^−11^ cm^2^ s^−1^ at 2.57 V, and the smallest $D_{Li^{+}}$ is 9.51 × 10^−11^ cm^2^ s^−1^ at 4.36 V. The calculated values are uniformly dispersed and consistent with the reported conversion cathode materials (Appendix A) [48,49,50,51].

Figure 6a shows the charge/discharge schematic process of the lithium-ion full-cell system assembled with the FeF_3_@C cathode and the prelithiation of the graphite anode [52]. To more effectively match the full cell, the constant current charge-discharge curve of the half-cell system with FeF_3_@C cathode and graphite anode was tested, as shown in Figure 6b. In the voltage window of 2.0–4.2 V, the full cell exhibits a high discharge capacity of 207.8 mAh g^−1^ and a charge capacity of 216.1 mAh g^−1^ at a current density of 0.4 C (Figure 6c), and the discharge capacity can maintain 154.3 mAh g^−1^ at the 40th cycle (Figure 6d). On the other hand, the energy density of the full cell is calculated to be 238.36 Wh kg^−1^ which is much higher than the commercial full-cell energy density of the traditional intercalation [53]. In addition, to validate the practicality of FeF_3_@C in a full cell, the “FeF_3_” pattern composed of 20 LED lamps is powered by the battery and has excellent luminescence, as shown in Figure 6e.

The FeF_3_ is also considered a conversion-type cathode material for Na^+^ storage. It is generally considered to undergo the following single electron transfer electrochemical reaction in the high voltage range (2.0–4.5 V) [28,54,55,56]:(6)$$FeF_{3}+Na^{+}+e^{-}↔NaFeF_{3}$$

To explore the storage mechanism of FeF_3_@C SIBs, the assembly and electrochemical performance measurements of SIBs were carried out. The rate performance of FeF_3_@C SIBs is depicted in Figure 7a,b. It is revealed that the specific capacities are 113.5, 102.7, 86.1, 74.7 and 67.9 mAh g^−1^ at the current rate of 0.2 C, 0.4 C, 0.6 C, 0.8 C and 1.0 C, respectively. When the current rate is back at 0.4 C, the capacity of the cell remains 93.4 mAh g^−1^ in the same voltage range. Compared with the impedance of FeF_3_@C after the first cycle and 100 cycles in Figure 7c, it is found that the radius of the half-circle increased significantly, and the slope of the inclined line in the low-frequency region decreased after the long cycle, indicating that the impedance increases slightly and the ion diffusion rate decreases, which may be related to the CEI film formed by the decomposition of the electrolyte. The SIBs can still maintain the capacity of 78.1 mAh g^−1^ after 100 cycles at a current density of 0.2 C (Figure 7d).

We also analyzed the storage mechanism of FeF_3_@C-SIBs by dynamics. The two pairs of redox peaks (P_1O_/P_1R_, P_2O_/P_2R_) in the CV curve of FeF_3_@C SIBs are more obvious as the scanning rate increases and the potentials are 3.14 V/2.78 V and 3.58 V/3.21 V, respectively in Figure 8a. Therefore, we speculate that the intermediate phase of Na_0.5_FeF_3_ is formed in the voltage window of 2.0–4.5 V. Similarly, the b-value is fitted according to the peak current, and the results show that the capacitance behavior is dominant in SIBs (Figure 8b and Appendix A). Significantly, the electrochemical behavior dominated by capacitance is proved by the kinetic analysis of Na^+^ storage, with a capacitive percentage of 58.6–76.2% at the scan rate of 0.4–2.0 mV s^−1^ (Figure 8c,d). When the SIBs are charged and discharged at 0.2 C current density, the GITT test curve of the second cycle is obtained in the range of 2.0–4.5 V (Figure 8e and Appendix A), and the $D_{Na^{+}}$ at different potentials is calculated. In Figure 8f, the $Log\;D_{Na^{+}}$ is between −11 and −13, showing smaller values than LIBs, which may be related to the larger radius of Na^+^. Appendix A shows that the $D_{Na^{+}}$ calculated in this work is greater than the average value of $D_{Na^{+}}$ in other relevant TMFs.

## 4. Conclusions

In summary, the FeF_3_@C nanocomposites with honeycomb mesoporous structures were prepared by gas fluorination. FeF_3_@C cathode was found to be suitable for the storage of alkali metal ions (Li^+^ and Na^+^). The capacity retention rate of LIBs after 100 cycles was up to 98.1%, and the capacity of SIBs reached 78.1 mAh g^−1^ at the current density of 1.0 C. At the same time, the FeF_3_@C cathode showed high capacity in full cells. Furthermore, through the dynamic analysis, it was found that the storage process of Li^+^ and Na^+^ in the electrode can be realized by surface pseudocapacitive behaviors and reversible redox, and the contribution proportion of pseudocapacitive increases as the current density increases. More significantly, this work enriches the in-depth understanding of the alkali metal storage mechanism in metal fluoride cathode materials, which can inspire more architecture designs for conversion-type materials in the future.

## Figures and Tables

**Figure 1 nanomaterials-12-04041-f001:**
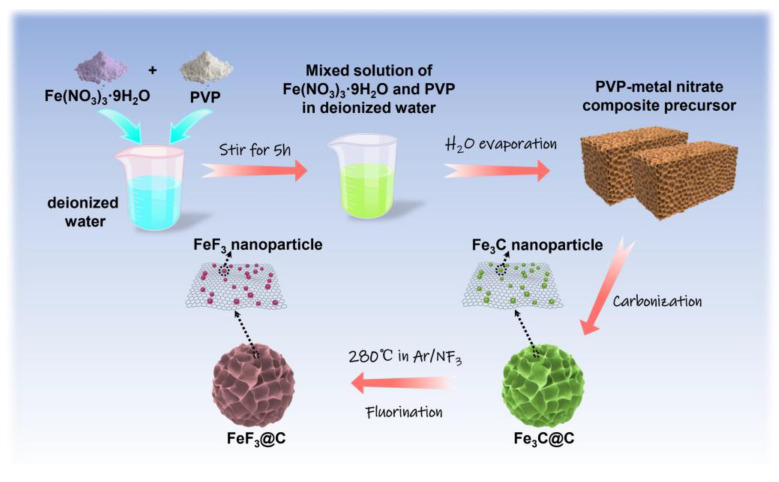
Schematic illustration of the synthetic process of the FeF_3_@C assemblies.

**Figure 2 nanomaterials-12-04041-f002:**
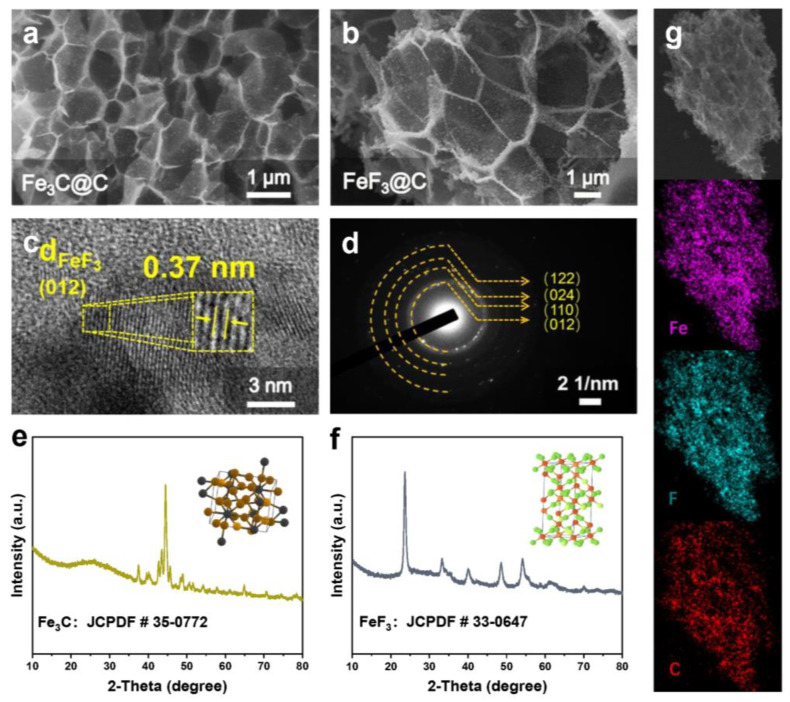
(**a**) FESEM of Fe_3_C@C and (**b**) FeF_3_@C, (**c**) HRTEM image and (**d**) SAED pattern of FeF_3_@C, (**e**) XRD pattern of Fe_3_C@C and (**f**) FeF_3_@C, (**g**) EDS mapping images of FeF_3_@C.

**Figure 3 nanomaterials-12-04041-f003:**
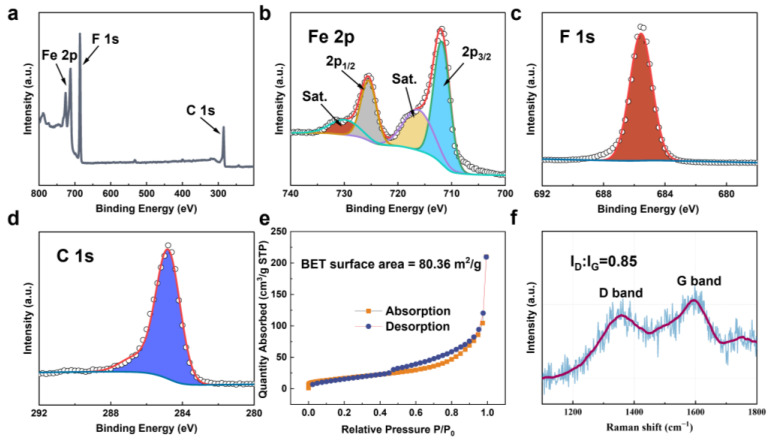
(**a**) XPS survey spectrum of FeF_3_@C and high-resolution of (**b**) Fe 2p, (**c**) F 1s and (**d**) C 1s. The BET surface area (**e**) and Raman spectra (**f**) of the obtained FeF_3_@C composites.

**Figure 4 nanomaterials-12-04041-f004:**
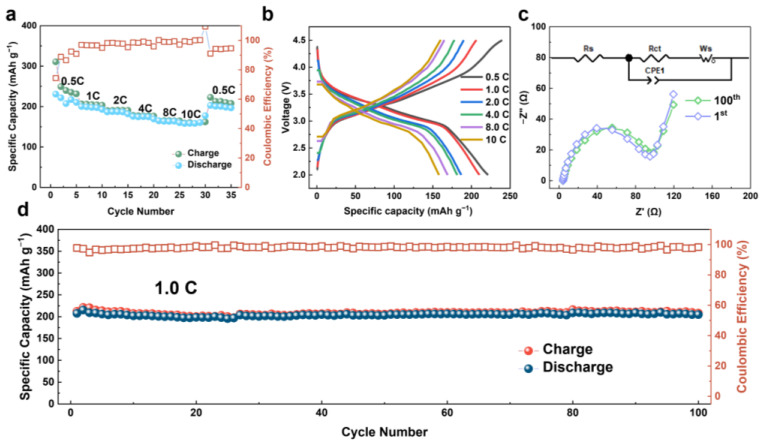
Electrochemical performance of FeF_3_@C-LIBs: (**a**) Discharge/charge capacities at different rates; (**b**) Discharge/charge profiles at corresponding rates; (**c**) Comparison of impedance spectra for 1 and 100 cycles of FeF_3_@C LIBs at open-circuit voltage; (**d**) Long-term cycle stability of the FeF_3_@C cathode at 1.0 C.

**Figure 5 nanomaterials-12-04041-f005:**
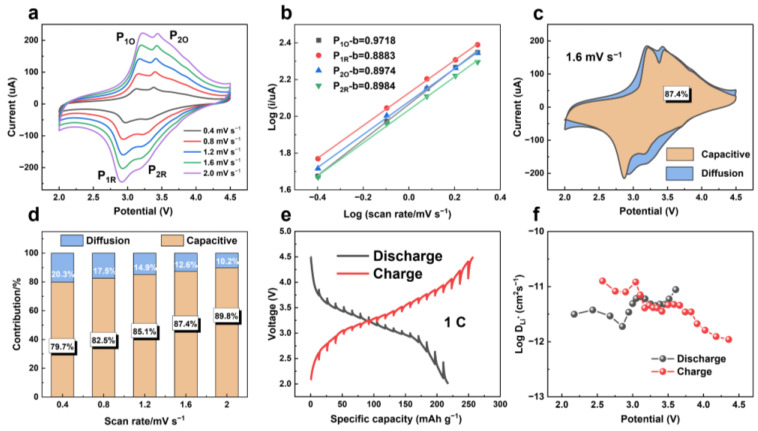
Kinetic features of Li^+^ storage mechanism in FeF_3_@C LIBs. (**a**) CV curves at the sweep rates of 0.4–2.0 mV s^−1^. (**b**) Determination of the b-value by the relationship between the characteristic peak current and different scan rates. (**c**) Percentage contribution of capacitive and diffusion to capacity in FeF_3_@C LIBs at 1.6 mV s^−1^. (**d**) Separation of capacitive contributions and diffusion-controlled contributions at different rates. (**e**) GITT charge-discharge curve of the second cycle at 1.0 C of FeF_3_@C LIBs. (**f**) Calculated Li^+^ diffusion coefficients at different voltages.

**Figure 6 nanomaterials-12-04041-f006:**
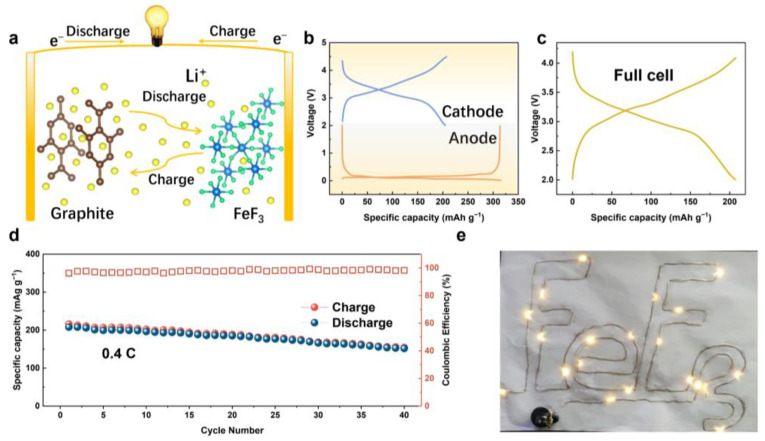
Electrochemical performance of full cell system coupling FeF_3_@C as cathode material and graphite as anode material. (**a**) Schematic showing the charge/discharge process. (**b**) The galvanostatic charge/discharge curves versus the specific capacity of FeF_3_@C cathode and a graphite anode. (**c**) The galvanostatic charge/discharge curves versus the specific capacity of the full cell. (**d**) Cycling performance during 40 cycles at 0.4 C. (**e**) Shows a photograph of the LEDs lit up by the full cell.

**Figure 7 nanomaterials-12-04041-f007:**
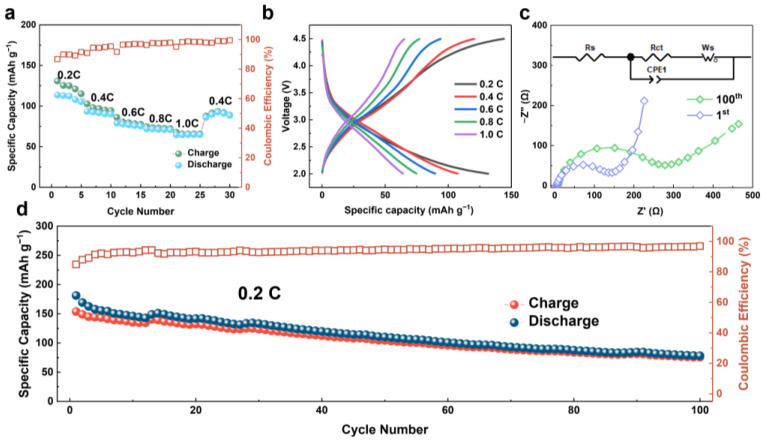
Electrochemical performance of FeF_3_@C-SIBs: (**a**) Discharge/charge capacities at different rates; (**b**) Discharge/charge profiles at corresponding rates; (**c**) Comparison of impedance spectra for 1 and 100 cycles of FeF_3_@C SIBs at open-circuit voltage; (**d**) Long-term cycle stability of the FeF_3_@C cathode at 0.2 C.

**Figure 8 nanomaterials-12-04041-f008:**
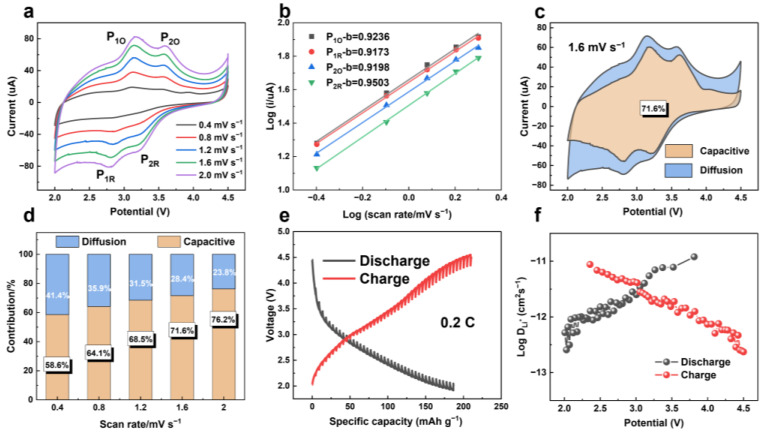
Kinetic features of the FeF_3_@C Na+ storage mechanism. (**a**) CV curves at the sweep rates of 0.4–2.0 mV s^−1^. (**b**) Determination of the b-value by the relationship between the characteristic peak current and different scan rates. (**c**) Percentage contribution of capacitive and diffusion to capacity in FeF3@C SIBs at 1.6 mV s^−1^. (**d**) Separation of capacitive contributions and diffusion-controlled contributions at different rates. (**e**) GITT charge-discharge curve of the second cycle at 0.2 C of FeF3@C SIBs. (**f**) Calculated Li+ diffusion coefficients at a different voltage.

## Data Availability

Not applicable.

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
