# Peer review of "Pseudocapacitance-Enhanced Storage Kinetics of 3D Anhydrous Iron (III) Fluoride as a Cathode for Li/Na-Ion Batteries"

_nanomaterials, 2022, doi:10.3390/nano12224041_

Round 1

Reviewer 1 Report

Authors have reported the synthesis of FeF3 and explored it as a cathode material for LIBs and SIBs. It has shown better cathode for LIBs and all the reporting results are convincing. It can be accepted in its current form.

Author Response

We are very grateful that you can review our manuscript in your busy time. At the same time, we appreciate your evaluation and recognition of our reporting results.

Reviewer 2 Report

This work is well narrated and some of the figures need to be replaced with high-quality ones. I have a few concerns regarding this work which are listed below. 

  1. Although the work is focused on both Li and Na, most of the results are limited to the case of only Li. Either the authors have to change the title or need to explain the reason for this.
  2. Can the authors comment on all possible redox reactions of transition metal fluorides (TMFs) during sodiation and de-sodiation?
  3. It has been mentioned in the manuscript that the Fe3C nanoparticles are uniformly embedded on the thin carbon wall (Fig. S1) and the composites of FeF3@C still maintain the honeycomb morphology after fluorination. However, from Fig. 2e and 2f, it looks like the structure is bulk in nature. Can the authors explain this?
  4. Can the authors give more insights into the diffusion and conductivity of the prepared samples? Whether the prepared compound show superior conductivity? See https://doi.org/10.1021/acsami.8b11476 for some insights. 

Author Response

Comments and Suggestions for Authors

This work is well narrated and some of the figures need to be replaced with high-quality ones. I have a few concerns regarding this work which are listed below.

(1) Although the work is focused on both Li and Na, most of the results are limited to the case of only Li. Either the authors have to change the title or need to explain the reason for this.

Response:

Thanks for the valuable suggestion from the reviewer. As the reviewer suggested, Fig. 1, 4, 5, 6, 7, and 8 have been replaced with high-quality ones. Although there is no full battery test for sodium ion batteries in the manuscript, we have conducted a systematic study on the storage mechanism of sodium ions in FeF3@C. In addition, the application prospect of TMFs in sodium ion batteries is introduced in the introduction, so we feel that the title of the article could summarize the main research contents of the article.

(2) Can the authors comment on all possible redox reactions of transition metal fluorides (TMFs) during sodiation and de-sodiation?

Response:

We are very grateful to your comments for the manuscript. At present, FeF3/FeF2-SIBs are most widely studied in TMFs, so we reviewed the relevant literature and summarized their electrochemical reactions [Nano Energy (2014) 10, 295–304], [RSC Adv.,2015,5,38277], [Journal of Alloys and Compounds 689 (2016) 945e951]:

  • The charge-discharge reaction of FeF3:

Initial discharge-charge cycle (4.5≥E≥0.8 V):

Discharge:

Charge:

(x+3y/2=1)

Intermediate discharge-charge cycles (4.5≥E≥2 V):

Discharge:

Charge:

(x+3y/2=1)

Finally stable discharge-charge cycle (4.5≥E≥2 V):

  • The charge-discharge reaction of FeF2:

Discharge:

Charge:

According to the above reaction, we can conclude that iron fluoride as a conversion cathode can transfer more electrons than the traditional intercalated cathode, so it shows a greater energy density. In addition, the redox reaction of the converted material is less limited by the size of the sodium ion during the reaction, so the damage to the material during the sodiation and de-sodiation process is less than that of the embedded material in theory.

According to your opinion, we have supplemented the sodiation and de-sodiation equations of FeF3 in the high voltage range of 2.0-4.5 V in the manuscript.

Changes made:

We have revised the second sentence of the second paragraph on page 8:

The FeF3 is also considered as a conversion-type cathode material for Na+ storage. It is generally considered to undergo the following single electron transfer electrochemical reaction in the high voltage range (2.0-4.5 V): [28, 55-57]

(6)

(3) It has been mentioned in the manuscript that the Fe3C nanoparticles are uniformly embedded on the thin carbon wall (Fig. S1) and the composites of FeF3@C still maintain the honeycomb morphology after fluorination. However, from Fig. 2e and 2f, it looks like the structure is bulk in nature. Can the authors explain this?

Response:

We are very grateful to your comments for the manuscript. The nanoscale Fe3C and FeF3 particles are observed in the FESEM images of Fig. S1 and Fig. S2. In addition, the interplanar spacing in the TEM image also proves the presence of FeF3 particles. The XRD patterns in Fig.2e and 2f are used to characterize the species of Fe3C@C and FeF3@C, where the illustrations are the atomic-scale crystal structures of Fe3C and FeF3, which are consistent with the results reported previously. [Adv. Mater. 2019, 31, 1905146]

(4) Can the authors give more insights into the diffusion and conductivity of the prepared samples? Whether the prepared compound show superior conductivity? See https://doi.org/10.1021/acsami.8b11476 for some insights.

Response:

We are very grateful for your insights on material properties and sharing of the article. The literature you recommend give us a lot of inspiration. It is reliable to calculate the diffusion dynamics of materials by density functional theory (DFT). Although we are unable to supplement the calculated data in time due to time issues, the argument in the literature that the graphitization of the material can effectively enhance ion transport and electron conduction matches our Raman results in this paper. As the reviewer suggested, more insights into the diffusion and conductivity of the prepared samples as well as the corresponding reference have been added in the revised manuscript.

Changes made:

The last sentence in the Raman result description was revised as:

As shown in the image, the G peak is higher than the D peak, indicating that the material has a high graphitization degree, which greatly improves the ion transport and electronic conductivity of FeF3.[37]

Reviewer 3 Report

The review is attached

Author Response

Comments and Suggestions for Authors

This paper summarized the use of metal fluoride (TMF) conversion cathodes, with high energy density recognized as promising candidates for next-generation high-energy Li/Na-ion batteries (LIBs/SIBs). The kinetics revealed an ion-dependent charge storage mechanisms and exceptional long-durability cyclic performance. A minor revision is suggested before publication. The detailed comments are as below:

(1) More discussion on the methods using composites materials is necessary to support their discussion on the synergistic contribution of pseudocapacitive and reversible redox behavior (ca. J. Clean. Prod. 2022, 359, 131994; J. Electrochem. Soc. 158 (2011) A1094).

Response:

Thanks a lot for the comments. It is well accepted that the nanocomposites can produce point defects, shorten the Li+/Na+ diffusion distance thus delivers high conductivity and Li+/Na+ storage. [ca. J. Clean. Prod. 2022, 359, 131994; J. Electrochem. Soc. 158 (2011) A1094] As the reviewer suggested, more discussions on the methods using composites materials as well as the corresponding references have been added in the revised manuscript.

Changes made:

The penultimate sentence in the introduction was revised as:

According to previous studies, nanoscale and carbon composite design are very efficient strategies to mitigate these problems. Which can produce point defects, shorten the shorten the Li+/Na+ diffusion distance thus delivers high conductivity and Li+/Na+ storage. [29, 30]

(2) On page 7: Details of the calculation of the molar volume of the active material should be explained.

Response:

Thank you very much for your attention to the problem of FeF3@C molar volume (VM). Our calculation steps for VM FeF3@C are as follows [Adv. Mater. 2019, 31, 1905146]:

The content of FeF3 in FeF3@C is 76%.

As the reviewer suggested, the molar volume of the active material has been explained in the manuscript.

Changes made:

The first sentence of the third paragraph on seventh page was revised as:

where the  (g) is mass, the  (g mol-1) and (cm3 mol-1) are the molecular weight and molar volume of the active material, respectively, where the  is about 28.5 cm3 mol-1 could be calculated according to the content of FeF3 in FeF3@C.

(3) Fig. 8 and Fig. 5: Kinetics features should be further compared with Li-, and Na- ion cells from literature (ca. Renew. Sust. Energ. Rev., 2020, 131, 109968).

Response:

Thanks for the valuable suggestion. As recommended, we summarize the lithium and sodium ion diffusion coefficients of transition metal fluorides (TMFs) reported in recent years as shown in tables S1 and S2 below, and place them in supporting information to support the conclusions in the manuscript. On the other hand, thank you for sharing the literature [ca. Renew. Sust. Energ. Rev. 2020, 131, 109968]. Based on the description of Li+/Na+/Mg+ ion batteries in the literature, we have supplemented and revised the manuscript.

Table S1 DLi+ comparison of various fluoride electrodes

Method

Materials

Magnitude of DLi+ (cm2s-1)

Reference

GITT

FeF3/C

10-11~10-9

[S1]

GITT

CoF2/Fe2O3

10-13~10-12

[S2]

EIS

FeF3/C

10-16~10-15

[S3]

EIS

FeF3·0.33H2O

(nanoparticle)

10-12~10-11

[S4]

EIS

FeF3·0.33H2O

(MOF)

10-15~10-14

[S5]

EIS

FeF3/graphitic carbon

10-15~10-14

[S6]

GITT

FeF3@C

10-12~10-11

This work

Table S2 DNa+ comparison of various fluoride electrodes

Method

Materials

Magnitude of DNa+ (cm2s-1)

Reference

GITT

FeF3·0.33H2O/rGO

10-14~10-12

[S7]

GITT

FeF3/graphene

10-13~10-11

[S8]

EIS

FeF3·0.33H2O@3D-OMCs

10-15~10-14

[S9]

EIS

FeF2@NGC

10-16~10-14

[S10]

GITT

FeF3@C

10-13~10-11

This work

Changes made:

The end of the fourth paragraph of the seventh page in the manuscript was revised as:

The calculated values are uniformly dispersed and consistent with the reported conversion cathode materials (Table S1).

The end of the first paragraph of the ninth page in the manuscript was revised as:

In Fig 8f, the  is between -11 and -13, showing smaller values than LIBs, which may be related to the larger radius of Na+. Table S2 shows that the  calculated in this work is greater than the average value of  in other relevant TMFs.

The fourth sentence in the introduction has been revised as:

Recently, sodium-ion batteries (SIBs) have been considered the most powerful competitor to replace LIBs.[7, 8]

(4) It is recommended to give the energy density (Wh/kg) currently attainable by Li- and Na-ion full cells in a new section of energy density and compare with some recent literature in similar systems (ACS Appl. Mater. Interfaces, 2022, 14, 43127).

Response:

Thank you very much for your recommendation. The literature you provide is very helpful to the manuscript. We calculated the energy density of the assembled FeF3@C/graphite lithium-ion full battery. The calculation process is as follows:

(207.8 mAh g-1·1.2×10-3 g×3,25 V)/3.4×10-3 kg=238.36 Wh kg-1

We have compared the results with other similar systems and the literature (ACS Appl. Mater. Interfaces, 2022, 14, 43127) has been cited in the manuscript.

Changes made:

The last sentence of the seventh page was revised as:

In the voltage window of 2.0-4.2V, the full cell exhibits a high discharge capacity of 207.8 mAh g-1 and a charge capacity of 216.1 mAh g-1 at a current density of 0.4C (Fig 6c), and the discharge capacity can maintain 154.3 mAh g-1 at the 40th cycle (Fig 6d). On the other hand, the energy density of the full cell is calculated to be 238.36 Wh kg-1 which is much higher than the commercial full cell energy density of the traditional intercalation.[54]
